# The Role of CO$_2$ as a Mild Oxidant in Oxidation and Dehydrogenation over Catalysts: A Review

**Sheikh Tareq Rahman [1]**, **Jang-Rak Choi [1,2]**, **Jong-Hoon Lee [1] and Soo-Jin Park [1,*]**

[1]  Department of Chemistry, Inha University, 100 Inharo, Incheon 22212, Korea; rahman19@inha.edu (S.T.R.); 22161120@inha.edu (J.-R.C.); boy834@naver.com (J.-H.L.)
[2]  Evertech Enterprise Co. Ltd., Dongtansandan 2 gil, Hwaseong 18487, Korea
*  Correspondence: sjpark@inha.ac.kr; Tel.: +82-32-876-7234

**Abstract:** Carbon dioxide (CO$_2$) is widely used as an enhancer for industrial applications, enabling the economical and energy-efficient synthesis of a wide variety of chemicals and reducing the CO$_2$ levels in the environment. CO$_2$ has been used as an enhancer in a catalytic system which has revived the exploitation of energy-extensive reactions and carry chemical products. CO$_2$ oxidative dehydrogenation is a greener alternative to the classical dehydrogenation method. The availability, cost, safety, and soft oxidizing properties of CO$_2$, with the assistance of appropriate catalysts at an industrial scale, can lead to breakthroughs in the pharmaceutical, polymer, and fuel industries. Thus, in this review, we focus on several applications of CO$_2$ in oxidation and oxidative dehydrogenation systems. These processes and catalytic technologies can reduce the cost of utilizing CO$_2$ in chemical and fuel production, which may lead to commercial applications in the imminent future.

**Keywords:** carbon dioxide; soft oxidant; oxidation; dehydrogenation; nano-catalyst

---

## 1. Introduction

Global warming is an imminent threat to our planet. It is essential to diminish the emission of greenhouse gases, especially carbon dioxide (CO$_2$), to slow global warming. Different sources of CO$_2$ emissions are a significant part to dictate by the ignition of liquid, solid, and gaseous chemicals. Rising atmospheric CO$_2$ concentrations and the increasing temperature of the planet's surface have increased public awareness of this problem [1]. CO$_2$ is utilized in the in the manufacturing industries, which is mostly released by the combustion of fossil fuels [2]. Among different products, methanol and formic acid can be synthesized from CO$_2$ which is used directly as fuels or to generate H$_2$ on demand at low temperatures (<100 °C) [1]. However, CO$_2$ can be used efficiently in various value-adding strategies and research pursuits which are converted waste emissions into valuable chemicals products, such as hydrocarbons and oxygenates [3]. Electrochemical activation technologies and conversion of CO$_2$ and H$_2$O into hydrocarbons has seen a marked increase in research activity over the past few years [4]. The impressive separation and utilization of CO$_2$ technologies in a higher challenge of organizing than other gases [5]. The development of CO$_2$ for novel approaches can add value to CO$_2$ recycling as it may result in commercially useful carbon-based products. Today, CO$_2$ is used commercially in the production of pharmaceuticals, air-conditioning systems, beverages, fertilizers, inert agents for food packaging, the water treatment process, fire extinguishers, and other applications. To achieve sustainable economic growth, it is crucial to study the conversion of CO$_2$ into carbon-based chemicals and materials. Industrial companies use massive amounts of CO$_2$ to enhance oil restoration. Biomass conversion to fuels also utilizes CO$_2$ [6]. Recently, Drisdell et al. [7] reported that oxide-derived copper catalysts are better at making fuel products from CO$_2$. According to Drisdell group, CO$_2$ is initially converted into carbon monoxide under first conditions for producing fuel and then hydrocarbon chains

are developed. Oxide-derived catalysts are better, not because they have oxygen remaining while they reduce carbon monoxide, but because the process of removing the oxygen creates a metallic copper structure that is better at forming ethylene. Using solar energy to convert $CO_2$ into most needed fuels has the potential to decrease global warming impact (GWI) and produce sustainable fuels at large scale [8]. A great deal of research has focused on combining heteroatoms in the carbon structure to improve the exchangeable action of $CO_2$ along with the adsorbent surfaces over the past few years [6].

CO_2 utilization has recently become an alluring sector of research, as it will help to alleviate climate change and reduce industrial operating costs. Globally, $CO_2$ capture and utilization are significant goals for chemicals and materials scientists [9]. Researchers are working to diminish the negative effects of $CO_2$ by adsorption [10,11], reduction, and fixation as well as through the development of metal-organic frameworks (MOFs), zeolites, polymers and micro-porous carbons [12]. Currently, $CO_2$ is used in an impenetrable phase under harsh conditions as an active promoter, making it a green substitute for organic compounds [13]. There are several limitations of dense phase $CO_2$ media, including the high pressures required to assure sufficient solubility of various transition metal catalysts and low reaction rates [14]. Jessop et al. proposed, as a solution to the solubility issue, an exchangeable process using 1,8-diazabicyclo-[5.4.0]-undec-7-ene. Additionally, they were able to eliminate partition steps by adjusting polarities with the use of $CO_2$ [15]. Another way to utilize $CO_2$ is to use it as an oxygen source. Park et al. demonstrated the mild oxidant character of $CO_2$ in the oxidative dehydrogenation of various types of alkyl benzene in both liquid and gaseous phases [16,17]. Using $CO_2$ in catalytic reactions offers other advantages; for instance, absorption of hydrogen from alkanes, alkyl aromatics, and alcohols using $CO_2$ as a reactant to create CO and oxygen species results in an expedited reaction rate, increased conversion, higher yield, and suppression of oxidation [16]. The presence of both $CO_2$ and $O_2$ increases the reaction rates as well as the conversion and selectivity. This process is performed under subcritical pressures of $CO_2$ and involves $CO_2$-promoted systems (CPS) instead of a $CO_2$-expanded system, as evidenced by the low-pressure approach as well as catalytic $CO_2$ activation. Recently developed $CO_2$ use technologies require the utilization of high-energy initiators [18]. Although great progress has been made in the carbon dioxide sector, there remain innate limitations, such as high-energy requirements, and the hydrogen recession. $CO_2$ has various benefits as a mild oxidant over several oxidizing promoters tested for oxidative dehydrogenation reaction, such as dry air, $SO_2$, and $N_2O$ [19]. C1 products such as methanol, formic acid has become possible to produce with high initial selectivity by using $CO_2$ over simple metal-based catalysts [4]. $CO_2$ promotes selectivity by contaminating the non-selective species of several catalysts, preventing the production of several by-products [20]. Additionally, $CO_2$ is used as a carbon source in the decoking process $(C + CO_2 = 2CO)$ which sustains catalytic activity [21]. Therefore, the oxidative dehydrogenation (ODH) reaction with $CO_2$ primarily considered to be a gas-interposed adaptation of the catalyst surface. This affects the diffusion, adsorption, and red-ox characteristics of the catalyst [22]. In recent years, the $CO_2$ conversion process has been utilized in various sectors, including thermo-chemical [23], photochemical [24], solar-chemical [25], electrochemical [26], biochemical [27] and homogenous catalysis [28] (Scheme 1).

In this review, we discuss a way to improve various technologies using $CO_2$ as a mild oxidant and enhancer for the production of essential chemicals. The purpose of this review is to illustrate the limitations and scope of $CO_2$ utilization and to highlight the advantages and challenges of carbon management. The use of $CO_2$ as a feedstock is a major goal, which could have a modest impact in practice, but may impart a significant symbolic effect on worldwide carbon stability. The further impact would result from the use of $CO_2$ as a soft oxidant and for oxidative dehydrogenation in catalytic reactions. Bartholomew et al. [29] studied the oxidizing capability of different gases in the gasification of coke. Their activities were ranked as follows: $O_2$ (105) > $H_2O$ (3) > $CO_2$ (1) > $H_2$ (0.003). This demonstrates that carbon dioxide is less active than molecular oxygen and water, but still offers high oxidative capacity. However, carbon dioxide has the greatest heat capability among the commonly used alternative gases. Furthermore, $CO_2$ can reduce the occurrence of hotspots, which cause problems, such as catalyst deactivation, runaway temperature, and undesirable product oxidation.

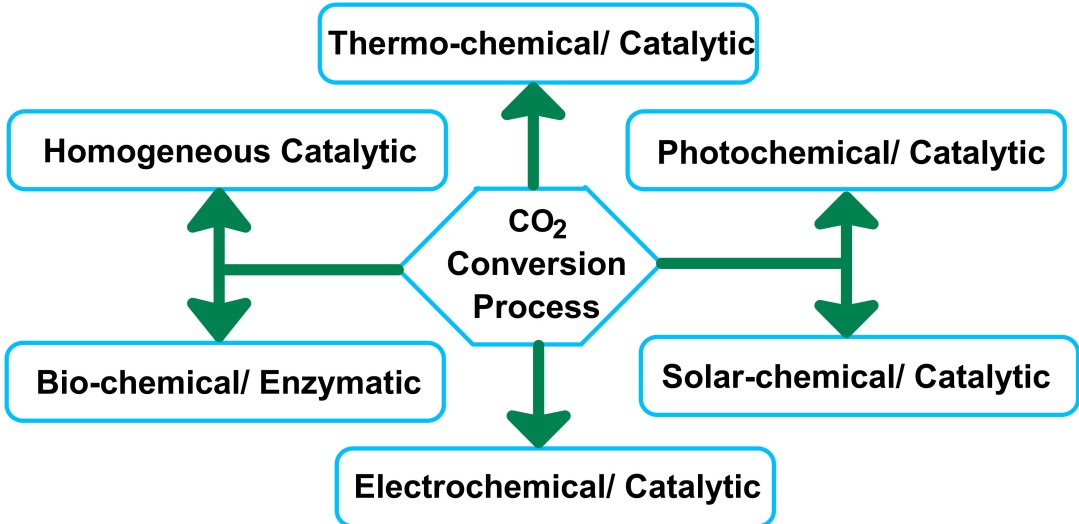

**Scheme 1.** The various chemical processes for $CO_2$ conversion.

## 2. Effect of $CO_2$ in Oxidation

### 2.1. Influence of $CO_2$ on Oxidation of Cyclohexene

The impact of $CO_2$, at various concentrations, was investigated on the oxidation of cyclohexene which is a small and symmetric molecule, similar to many starting compounds in chemical synthesis (Scheme 2) [30]. The results revealed that $O_2/CO_2$ conversion (%) was higher than $O_2/N_2$ conversion (%) rate. However, at a gas ratio of 0.066 $O_2$:$CO_2$/$N_2$ (Table 1, entry 1), cyclohexene was not converted. Park et al. revealed the positive impact of carbon dioxide on mesoporous metal-free oxidation carbon nitride (MCN) catalysts [31]. These mesoporous MCN elements exhibit oxygen-carrying capabilities which are effective sites for oxidation. Additionally, the large nitrogen quantity in the CN matrix acts as a $CO_2$-philic exterior for the incitation of $CO_2$. Molecular oxygen promotes this synergy, allowing for the oxidation of cyclic olefins and improving the conversion of cyclic olefins with better selectivity. In-between the conversion of the $O_2/CO_2$ and the $O_2/N_2$, Park et al. observed the enhancive performance as a premier time, which can be expressed as ΔC (%) and can be calculated using the Equation (1):

$$\Delta C(\%) = \frac{\left(C_{O_2/CO_2}\right) - \left(C_{O_2/N_2}\right)}{\left(C_{O_2/CO_2}\right) + \left(C_{O_2/N_2}\right)} \times 100 \tag{1}$$

where,

$$\left(C_{O_2/CO_2}\right) = \text{ Conversion in } O_2/CO_2 \text{ and } \left(C_{O_2/N_2}\right) = \text{ Conversion in } O_2/N_2$$

**Scheme 2.** Cyclohexene oxidation reaction over catalyst. (Redrawn from [30]; copyright (2018), WILEY-VCH). (**A**) = 2-cyclohexene-1-one, (**B**) = cyclohexene oxide, (**C**) = 2-cyclohexene-1-ol, (**D**) = 2-cyclohexene-1-hydroperoxide). Reaction conditions: 10 bar $O_2$; 2.5 mL cyclohexene; 0.5 mL cyclohexane(IS); 10 mg catalyst; 15 mL MeCN; stirred in an autoclave (1000 rpm); 70 °C; 16 h.

**Table 1.** Effect of the $CO_2$ on oxidation of cyclohexene over MCN Ref [31] (Reproduced from [31]; copyright (2011), Royal Society of Chemistry).

| Entry | Gas Ratio (PSI) [a] | Conversion (%) $O_2/CO_2$ | Conversion (%) $O_2/N_2$ | $\Delta C$(%) [b] |
|---|---|---|---|---|
| 1 | 0.066 | 0 | 0 | 0 |
| 2 | 0.142 | 16 | 9 | 28 |
| 3 | 0.230 | 25 | 18 | 16.3 |
| 4 | 0.333 | 33 | 24 | 15.7 |
| 5 | 0.454 | 34 | 24 | 15.7 |

Reaction conditions: 20 mg Melamine mesoporous carbon nitride (M-MCN), 10 mL Dimethylformamide (DMF), temperature 373 K, Pressure 80 PSI, time 10 h; Estimated by Gas Chromatography (GC) analysis. [a] PSI = Pounds per Square Inch, [b] Conversion (%) of cyclic olefin.

The efficiency of $CO_2$ in the oxidation of cyclohexene at varying $CO_2$ concentrations is shown in Table 1. Higher conversions were achieved by the $O_2/CO_2$ system. The results showed that the conversion of cyclohexene was nothing at a content of 0.066 $O_2$ (entry 1). This is Possibly due to the low frictional pressure of $O_2$, which is deficient to drive the reaction. Further, the $\Delta C$% value was higher for higher concentrations of $CO_2$. No meaningful change of $\Delta C$% was demonstrated for gas ratios beyond 0.333 in the catalytic process, demonstrating the impregnation of activity.

$CO_2$ has been used with metal-supported systems that were observed to produce a per-oxycarbonate species which are highly active in oxidation reactions. Aresta et al. were reported the composition of a metal per-oxycarbonate species, as determined by spectroscopic analysis [32]. A process for the production of per ox-carbonate has acceded in Scheme 3. Park et al. investigated the oxidation of alkyl aromatics via an EPR analysis using a metal carbonate catalyst. They demonstrated the production of metal per-oxycarbonate groups in the presence of carbon dioxide by the hyperfine cracking of manganese. Yoo et al. [20] observed the production of per-oxycarbonate on Fe/Mo/DBH (deboronated borosilicate molecular sieve); the production of per-oxycarbonate is illustrated in Figure 1. All of the catalytic schemes discussed above involve transitional metal catalysts and $CO_2$ coupled with oxygen. The resulting enhancement over traditional metal oxide systems in $O_2/CO_2$ mixtures may occur because of an oxygen exchange between $O_2$ and $CO_2$, which would increase the rate of the reaction. During isotope-labeling studies, these types of exchanges have been detected by Iwata et al. [33] using different metal oxide structures.

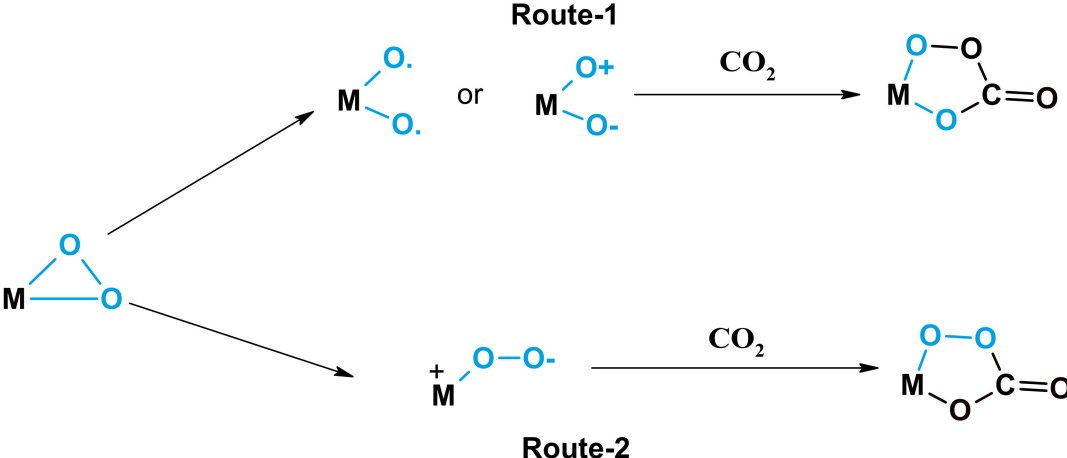

**Scheme 3.** Per-oxycarbonate production reaction mechanisms (Redrawn from [32]; copyright (1996), American Chemical Society).

**Figure 1.** Per ox-carbonate over Fe/Mo/DBH in the $O_2$/$CO_2$ system (Reproduced from [20]; copyright (1993), Elsevier (Amsterdam, The Netherlands)).

## 2.2. Promotional Effect of $CO_2$ on Oxidation of Cyclic Olefins

Park et al. demonstrated the use of $CO_2$ as a promoter for the oxidation of cyclic olefins with mesoporous carbon nitrides (CN) as a metal-free catalyst in the presence of molecular oxygen. Analysis of the surface characteristics of the catalyst after the reaction revealed the presence of carbamate, confirmed by a new band in the FTIR spectrum at 1419 cm$^{-1}$. This measurement illustrated the incitation of $CO_2$ owing to the accumulation of surface carbamate. This surface carbamate can then react with the cyclic olefins, assisted by the catalyst. After the reaction, the IR spectra showed the presence of extra bands at 2174 and 2115 cm$^{-1}$, possibly due to a gaseous CO doublet. However, these absorption bands were not present before the reaction. This analysis exposed the production of CO, which is revealed to the increased catalytic activity to credit to carbon dioxide sharing as an 'oxygen atom' onset [31,32]. The production of CO was previously observed in nitrogen including heterocyclic systems [34–36]. The positive impact of $CO_2$ in the oxidation of cyclic olefin was quantified by measuring the catalytic performance using various reactants, cyclopentene ($n = 1$), cyclohexene ($n = 2$), cyclooctene ($n = 4$), and cyclododecene ($n = 8$) (Table 2). The epoxide selectivity was greater in $O_2$/$CO_2$ than $O_2$/$N_2$, suggesting that in the presence of $CO_2$, the mechanism may be altered to improve the conversion and selectivity. The blend of gaseous from the autoclave was studied by IR spectroscopy to better understand the positive impact of $CO_2$. In the reaction with no oxidant and source oxygen, it was presumed that $CO_2$ is reduced to CO and aldehyde is oxidized to carboxylic acid in the same process. The reaction may have occurred via the addition of carbon dioxide to the quickly produced Breslow intermediate A to produce the hydroxy carboxylate B and the tautomer C (Scheme 4) [34]. Possibly, the following intermediate can lose CO and hydroxide to support benzoic acid. Additionally, it was observed that intermediate D is supplicated in the oxidative esterification of aldehydes with CO. Interestingly, phenylglyoxylic acid was revealed to nucleophilic heterocyclic carbenes (NHC) under similar experimental conditions wherein phenylglyoxylic acid was switched to benzoic acid. (Scheme 5). Under mild experimental conditions, $CO_2$ was utilized in an NHC-intermediated conversion of the aldehyde to the carboxylic acid.

**Table 2.** Enhancive role of $CO_2$ on cyclic olefins oxidation (Reproduced from [31]; copyright (2011), Royal Society of Chemistry).

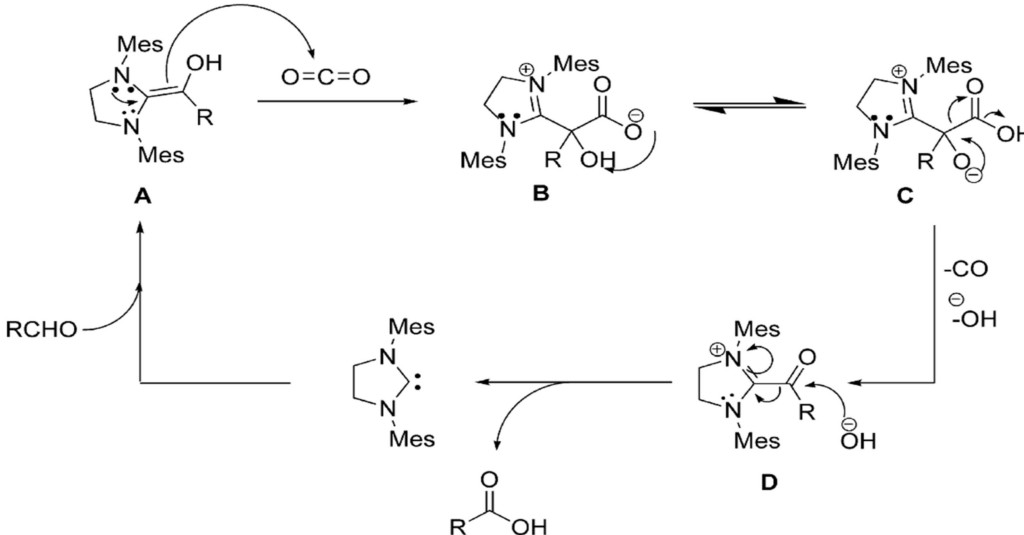

$O^C = O_2/CO_2$

$O = O_2/N_2$

| Entry | n | Gas | Conversion of 3 (%) | Selectivity (%) | | | |
|---|---|---|---|---|---|---|---|
| | | | | 4 | 5 | 6 | ΔC (%) |
| 1 | 1 | $O^C$ | 40 | 37 | 24 | 29 | 12.6 |
| | | O | 31 | 30 | 22 | 40 | - |
| 2 | 2 | $O^C$ | 33 | 30 | 21 | 49 | 15.8 |
| | | O | 24 | 25 | 16 | 53 | - |
| 3 | 4 | $O^C$ | 21 | > 99 | - | - | 27.0 |
| - | - | O | 12 | > 99 | - | - | - |
| 4 | 8 | $O^C$ | 17 | > 99 | - | - | 30.7 |
| - | - | O | 9 | > 99 | - | - | - |

Reaction conditions: 20 mg Melamine mesoporous carbon nitride (M-MCN), 10 mL Dimethylformamide (DMF), temperature 373 K, Pressure 80 PSI, gas ratio 0.333, time 10 h; Produced analyzed by GC and GC-MS.

**Scheme 4.** Proposed Mechanism for aldehyde assisted $CO_2$ to carboxylic acid process. (Reprinted from [34]; copyright (2010), American Chemical Society).

**Scheme 5.** Phenylglyoxylic acid to benzoic acid reaction with NHC-intermediate. (Reprinted from [34]; copyright (2010), American Chemical Society).

### 2.3. Influence of $CO_2$ on Oxidation of p-Xylene

It was proposed that in the $O_2$-$CO_2$ system, metal peroxy-carbonate groups assist as oxygen transfer promoters to the oxyphilic substrate. Aresta et al. [32] also reported that the presence of O-O bonds in Rh ($\eta^2$-$O_2$) complexes imply the accumulation of metal per-oxycarbonate during the other oxidation reaction. They demonstrated that $CO_2$ promotes the oxidative ability of $O_2$ over the RhCl(Pet$_2$-Ph)$_3$ catalyst. In the presence of $CO_2$ over the metal-based structure was found to be formation of peroxycarbonate species which are more active than hydrogen peroxide in oxidation reaction [37]. Park et al. [22] reported the performance of carbon dioxide in the liquid-phase oxidation reaction of toluene, *p*-tolu-aldehyde, and *p*-xylene with $O_2$ over an MC-based catalyst (Co/Mn/Br). The reaction rate, selectivity, and the conversion were all enhanced by the co-presence of $CO_2$. This enhancement was attributed to the creation of per-oxycarbonate species, as determined by electron paramagnetic resonance (EPR) analysis of the reaction with and without carbon dioxide. A hyperfine manganese arrangement was noticed in the existence of $CO_2$, confirming the formation of a per-oxycarbonate species.

Additionally, Park et al. observed the oxidation of different alkyl aromatics applying MC-supported catalysts [22]. Oxidations were carried out using $O_2$ as the oxidant (with $N_2$) and compared to reactions in the presence of both $O_2$ and $CO_2$. The conversion of *p*-xylene without $CO_2$ (Table 3) was 57.2%, whereas the conversion of *p*-xylene was increased to 66.8% in the presence of $CO_2$. Moreover, in $O_2$/$CO_2$, the yield of terephthalic acid was improved. The Amoco Chemical Research Laboratory studied the activation of $CO_2$ in the gas-state of *p*-xylene oxidation to *p*-tolualdehyde and terephthaldehyde over the chemical vapor deposition (CVD) of Fe/Mo/DBH [20]. The oxidation reaction was performed in two feed streams varying compositions, including *p*-xylene with $O_2$/$N_2$/He and *p*-xylene with $O_2$/$N_2$/$CO_2$. The catalytic activity is shown in both the feeds at various temperatures in Figure 2, as shown in the figure, *p*-xylene conversion in the existence of $CO_2$ in $O_2$ was greater than the absence of $CO_2$ in $O_2$. This improved conversion was connected to the production of per-oxycarbonate groups over the catalyst surface. Furthermore, in the existence of $CO_2$, the secondary reactions also emerged more remarkable, possibly due to the acidity of the $CO_2$ molecules adsorbed onto the DBH matrix. In comparison with $O_2$ alone, the conversion of *p*-xylene was higher in the co-presence of $CO_2$ at all temperatures (Figure 2). The $O_2$/$N_2$/$CO_2$ feed system, resulted in a higher conversion of *p*-xylene and greater selectivity towards benzaldehyde at temperatures from 300 °C to 375 °C (Table 4). It was observed that no carbon dioxide was formed by the burning of *p*-xylene over the catalyst at 375 °C; however, in the $O_2$/$N_2$/He feed system, the formation of $CO_2$ started (10.7%) at 300 °C and significantly increased (20.2%) at 375 °C. Thus, $CO_2$ performed as a co-oxidant for the gas-phase *p*-xylene oxidation reaction with oxygen. Yoo et al. [20] also reported a significant enhancement in the conversion of *p*-xylene, *p*-ethyl toluene, and *o*-xylene in the presence of $CO_2$ at varying temperatures.

**Table 3.** Effect of $CO_2$ on the MC-type catalyst for oxidation of *p*-xylene (Reproduced from [22]; copyright (2012), Royal Society of Chemistry).

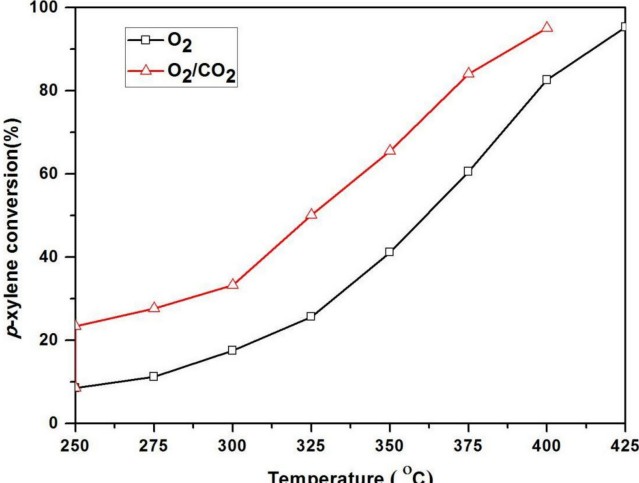

| Gas | Conversion of 1 (%) | Yield mol (%) | | | | |
|-----|---------------------|------|------|------|------|------|
| | | 2 | 3 | 4 | 5 | 6 |
| $O_2$ | 57.2 | 17.7 | 47.9 | 2.8 | 1.7 | 29.2 |
| $O_2/CO_2$ | 66.8 | 34.8 | 36.9 | 1.7 | 2.4 | 24.2 |

Reaction conditions: Temperature 170 °C, time 3 h, Mesoporus carbon (MC) type catalyst with transition metal additive (Co/Mn/Br), Co 100 ppm, Mn 200 ppm, Br 300 ppm [38].

**Figure 2.** Promotional role of $CO_2$ on Fe/Mo/DBH for the oxidation of *p*-xylene.

**Table 4.** Enhancive effect of $CO_2$ on oxidation of *p*-xylene (Reproduced from [20]; copyright (1993), Elsevier). Reaction conditions: WHSV: 0.22 h$^{-1}$, contact time: 0.21 s, gas flowrate: 400 sccm, Feed gas 1: 0.1% *p*-xylene, 1% $O_2$, 1% $N_2$ in He. Feed gas 2: 0.1% *p*-xylene, 1% $O_2$, 1% $N_2$ in commercial grade $CO_2$.

| Temperature (°C) Feed [a] | 300 | | 350 | | 375 | |
|---------------------------|------|------|------|------|------|------|
| | 1 | 2 | 1 | 2 | 1 | 2 |
| *p*-Xylene (Con.%) | 17.6 | 33.3 | 41.2 | 65.5 | 60.7 | 84.1 |
| Product selectivity (mol%) | - | - | - | - | - | - |
| *p*-Tolu-aldehyde | 57.9 | 57.2 | 50.2 | 40.9 | 40.6 | 40.6 |
| Terephthaldehyde | 16.4 | 27.5 | 23.5 | 33.6 | 32.6 | 30.2 |
| Benzaldehyde | 1.3 | 1.5 | 2.4 | 2.7 | 2.7 | 3.1 |
| Maleic anhydride | 0.0 | 0.0 | 2.4 | 6.0 | 5.8 | 13.7 |
| Toluene | 6.2 | 6.9 | 3.5 | 4.8 | 3.1 | 4.8 |
| Trimethyl biphenyl methane | 7.5 | 6.8 | 1.7 | 0.6 | 0.4 | 0.0 |
| CO | 0.0 | 0.0 | 0.6 | 3.4 | 4.7 | 7.5 |
| $CO_2$ | 10.7 | 0.0 | 15.6 | 0.0 | 20.2 | 0.0 |

[a] Feed gas 1: $O_2/N_2/He$, Feed gas 2: $O_2/N_2/CO_2$.

### 2.4. Oxidation of p-Toluic Acid and p-Methyl-Anisole

$CO_2$ acts as a promoter in catalytic systems and as a co-oxidant with $O_2$ resulting in improved reaction kinetics, more desirable product distributions, better selectivity, and higher conversion. Initially, Aresta et al. [32] reported that carbon dioxide enhanced the oxidative characteristics of dioxide in transition metal systems. Park et al. [38] studied the use of Co/Mn/Br catalysts in the fluid- phase oxidation of olefins. Interestingly, they observed the expansion effect of carbon dioxide on mesoporous carbon nitride (MCN) catalytic systems, whereas the $CO_2$-promoted system was fabricated by them on the oxidation of alkyl-aromatics. In the presence of $CO_2$, the conversion of *p*-toluic acid over the metal carbonate (MC) catalyst was increased by 12% (Table 5) compared to oxidation in $O_2$ alone. Furthermore, the yield of terephthalic acid increased from 58.2% to 64.9%. These data demonstrate that the catalytic activity is significantly enhanced by $CO_2$. Interestingly, over an MC-supported catalytic system, the main product of the oxidation of *p*-methyl-anisole is *p*-methoxy phenol (Table 6) along with a limited number of other products, such as *p*-anisaldehyde and *p*-anisic acid. However, the yield of *p*-anisaldehyde has increased the presence of $CO_2$, again demonstrating the capacity of $CO_2$ to sustain mono-oxygen transfer.

**Table 5.** *p*-toluic acid oxidation with $CO_2$ on an MC-type catalyst (Reproduced from [22]; copyright (2012), Royal Society of Chemistry).

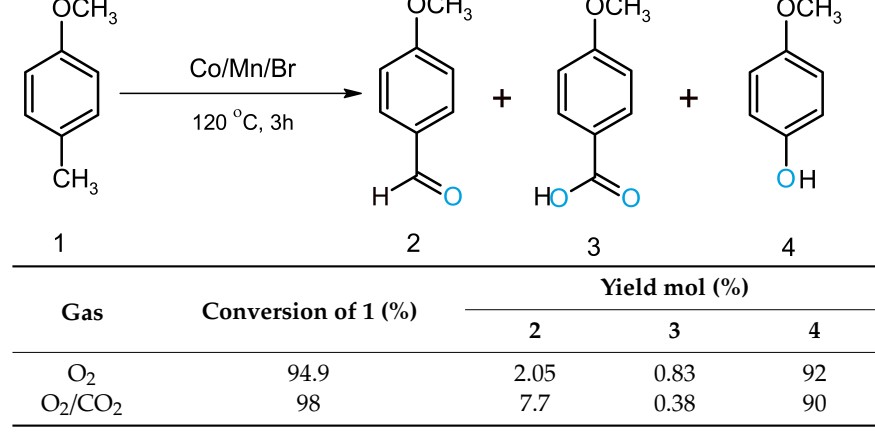

| Gas | Conversion of 1 (1%) | Yield mol (%) | |
| --- | --- | --- | --- |
| | | 2 | 3 |
| $O_2$ | 60.9 | 58.2 | 3.7 |
| $O_2/CO_2$ | 72.7 | 64.9 | 10.6 |

Reaction conditions: 6 mL *p*-toluic acid, 0.1183 g $CoBr_2$, 0.1587 g $Mn(OAc)_2\cdot4H_2O$ in 24 mL HOAc, temperature 190 °C, time 3 h; $P_{CO2}$ = 0-6 atm, $P_{O2}$ = 2 atm [38].

**Table 6.** Performance of $CO_2$ on oxidation of *p*-methylanisole (Reproduced from [22]; copyright (2012), Royal Society of Chemistry).

| Gas | Conversion of 1 (%) | Yield mol (%) | | |
| --- | --- | --- | --- | --- |
| | | 2 | 3 | 4 |
| $O_2$ | 94.9 | 2.05 | 0.83 | 92 |
| $O_2/CO_2$ | 98 | 7.7 | 0.38 | 90 |

Reaction conditions: 43.5 mmol *p*-methylanisole, 0.6 mmol $CoBr_2$, 0.6 mmol $Co(OAc)_2$, 0.6 mmol $Mn(OAc)_2\cdot4H_2O$ in 30 g HOAc, total pressure 12 atm ($P_{CO2}$ = 0-2 atm, $P_{O2}$ = 2,3,6 atm, $P_{N2}$ balance). temperature 120 °C, time 3 h [38].

## 3. Performance of $CO_2$ in Oxidative Dehydrogenation

### 3.1. Influence of $CO_2$ on Dehydrogenation of Ethyl Benzene

Styrene is typically formed by the dehydrogenation of ethyl benzene under the steam on a metal oxide catalyst in an adiabatic reactor [39]. There are several limitations to this process, including thermodynamic drawbacks, low conversion rates, high endothermic energy ($\Delta H^o_{298}$ = 123.6 kJ mol$^{-1}$), huge energy destruction, and catalyst deactivation by coke production [40]. An alternative method of styrene production is the oxidative dehydrogenation reaction with $O_2$; however, this results in the burning of large quantities of valuable hydrocarbons. In this context, the use of $CO_2$ in the oxidative dehydrogenation of ethyl benzene may prove useful [39–61]. *Zhang* et al. [43] confirmed coke deposition using spectroscopy and reported the deactivation of a ceria catalyst without $CO_2$ present. In a two-step, reaction mechanism for the dehydrogenation of ethyl benzene to produce styrene with $H_2$ in the initial step and in the presence of $CO_2$, ejection of $H_2$ through a reverse water-gas shift (RWGS) reaction was also demonstrated [41]. Kovacevic et al. revealed the results of $CeO_2$ catalyst morphology (i.e., rods vs. cubes vs. particles) in the presence and absence of $CO_2$ [42]. They reported that in the presence of $CO_2$ cubic catalysts showed higher initial benzene selectivity, and about two times more activity per m$^2$ compared to the reaction without $CO_2$. Interestingly, the number of oxygen species was increased by the presence of $CO_2$. They also observed that these additional oxygen molecules were expended in the ethyl benzene conversion, demonstrating their performance as active sites for styrene formation. Periyasamy et al. reported that in the ODH reaction the conversion of ethyl benzene (EB) was 50% and the selectivity for styrene was 93% at gas hourly space velocity (GHSV) 2400 h$^{-1}$. They also observed that the conversion and selectivity increased with enhancing oxidant flow ratio, up to GHSV 2400 h$^{-1}$.

Park et al. [49] reported on the use of SBA-15 as a beneficial backing for a ceria-zirconium (25:75) combined oxide catalyst for oxidative dehydrogenation of ethyl benzene utilizing carbon dioxide. Ce-Mn oxide nanoparticles enclosed inside carbon nanotubes (CNTs) were used for the oxidative dehydrogenation of ethyl benzene with $CO_2$ acting as a soft oxidant. The high diffusion and the encapsulation effect of CNTs resulted in excellent performance of the entrapped catalysts. Correlated to $CeO_2$ support CNTs, the restriction result of CNT pathways enhanced the communication between carbon nanotube (CNT) inner walls and $CeO_2$ particles, which is orderly, convinced the misrepresentation of $CeO_2$ crystal lattice which is advertised $CeO_2$ reduction and invigoration of $CeO_2$ surface lattice oxygen. The unique process of promoting oxidative catalytic activity the addition of $CO_2$ was reported by Zhang et al. [44]. They observed that multi-walled carbon nanotubes (MWCNTs) have a significant quantity of surface hydroxyl groups which are produced by an alkali-supported hydrothermal method after ball milling. The MWCNTs can mostly arrange the active sites for the oxidative dehydrogenation of ethyl benzene (EB) in the existence of $CO_2$. Figure 3a shows the conversion of ethyl benzene over various types of MWCNTs at 3 hr. The HMWCNTs-OH exhibits significant catalytic activity, indicating that the surface hydroxyl groups are the active sites for the oxidative dehydrogenation of ethyl benzene. The O1s spectra of HMWCNTs-B-OH identified by XPS is shown in Figure 3b. Figure 3c demonstrates the production of carbonyl groups in the reaction. The results indicate that $CO_2$ acts effectively as a soft oxidant, directly oxidizing -OH groups into carbonyl groups. As shown in Figure 3d, CO and $H_2$ were also identified as byproducts for the reaction, indicating that $CO_2$ is reduced in the RWGS reaction. $CO_2$ activation occurs via electron donation from the surface of the catalyst to the anti-bonding orbital of $CO_2$ [62]. However, ethyl benzene (EB) can be activated for oxidative dehydrogenation (ODH) by donating an electron to the acidic portion of the catalyst surface.

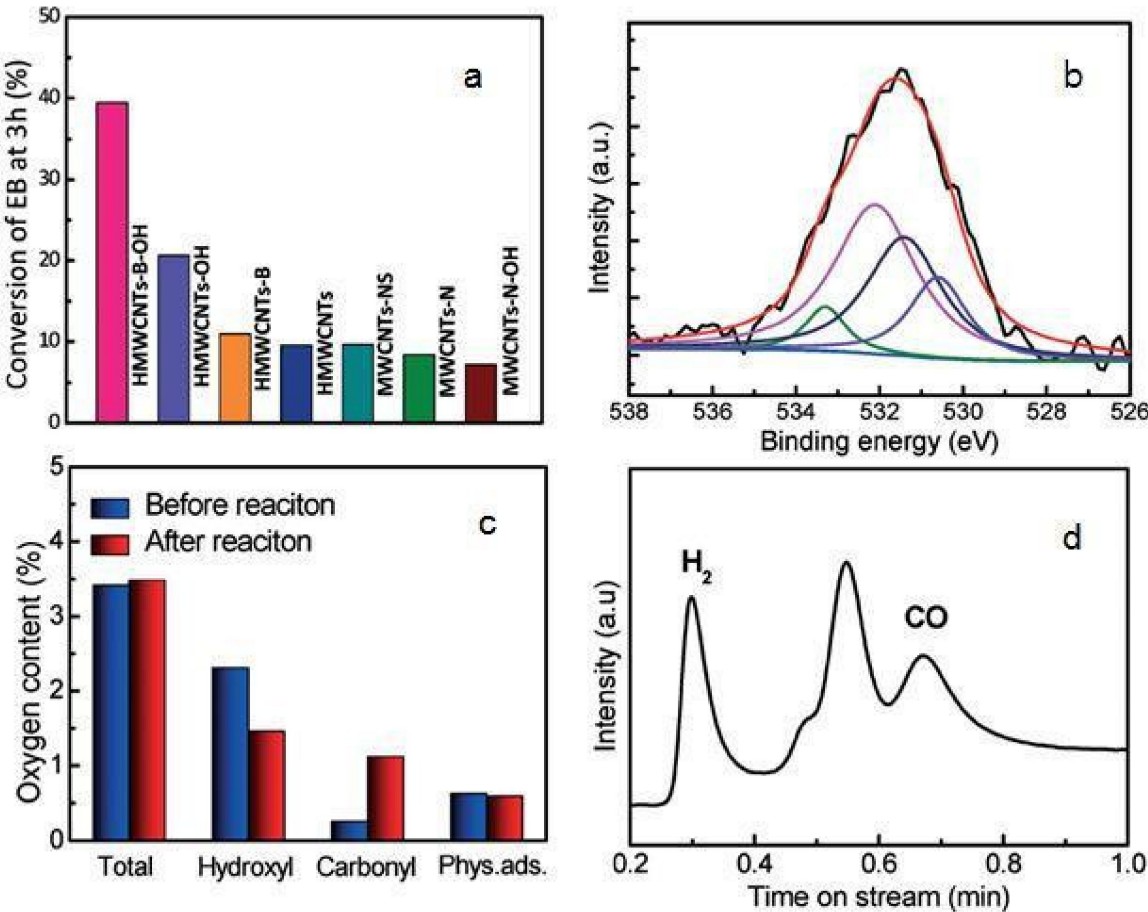

**Figure 3.** (**a**) Conversion of EB using $CO_2$ as a soft oxidant; (**b**) O1s spectra of HMWCNTs-B-OH after oxidative dehydrogenation by XPS for 3 hr; (**c**) Oxygen substance (mol%) of HMWCNTs-B-OH earlier and later oxidative dehydrogenation reaction at 3 h; (**d**) Gas derivatives of HMWCNTs-B-OH after oxidative dehydrogenation at 3 h. (Reprinted from [44]; copyright (2013), Royal Society of Chemistry).

Additionally, basic sites abstract hydrogen from ethyl benzene. Thus, the aggregated effect of the basic and acidic sites of the catalyst face is the oxidative dehydrogenation reaction, resulting in high catalytic efficiency in the existence of $CO_2$ [63]. Sato et al. reported on the use of $CO_2$ as a mild oxidant in the oxidative dehydrogenation reaction as well as the typical dehydrogenation process in the absence of $CO_2$. Two mechanisms utilizing acidic and basic sites were proposed, as depicted in Figure 4. Vanadium-embed catalysts also used in $CO_2$ based oxidative dehydrogenation of oxidative dehydrogenation of ethyl benzene (ODHEB) reactions [21,45]. $CO_2$, being a mild oxidant, cannot reproduce the active sites on the $V_2O_5$ (001) surface of the catalyst quickly enough due to the large activation energy (3.16 eV) [46]. A ceria-supported vanadium catalyst floated on a titania-zirconia combined oxide ($TiO_2$-$ZrO_2$) has moderate constancy which was reported by Reddy et al. [47]. XPS analysis of Ce 3d indicated the presence of $Ce^{4+}$ and $Ce^{3+}$ on the Ti-Zr catalyst. They also reported that $CeO_2$-$V_2O_5$/$TiO_2$-$ZrO_2$ (TZ) catalysts resulted in 56% conversion of ethylbenzene and 98% selectivity of styrene. Liu et al. [48] illustrated the red-ox mechanism for the $CO_2$-oxidative dehydrogenation of ethyl benzene ($CO_2$-ODEB) using a ceria promoted vanadium catalyst, as shown in Figure 5. In the $CO_2$-ODEB case, $CO_2$ directly oxidizes $Ce^{3+}$ to $Ce^{4+}$, and ethylbenzene reduces of $V^{5+}$ to $V^{4+}$. Then, the reduction of $Ce^{4+}$ to $Ce^{3+}$ and the oxidation of $V^{4+}$ to $V^{5+}$ completes the full cycle. In the existence of $CO_2$, modified vanadium catalysts are effective, selective, and stable for the ODEB, as reported by Park et al. [49] Rapid regeneration of active sites on a silica-assisted vanadium catalyst along in the presence of $CO_2$ has also been reported [64]. 10% $La_2O_3$-15%$V_2O_5$/SBA-15 (wt.%)

catalyst resulted in a 74% styrene yield, with La$^{3+}$ resisting coke ejection [50]. The use of supporting materials, such as Aluminum mesoporous cylindrical molecular sieve (Al MCM-41) also resulted in substantial EB conversion in the ODEB using a VO$_x$/Al MCM-41 catalyst in the presence of CO$_2$ [51]. ZrO$_2$-containing combined oxide catalysts for oxidative dehydrogenation of ethylbenzene with CO$_2$ in the presence of MnO$_2$, CeO$_2$ and TiO$_2$ have exhibited high activity. The styrene yield was also increased over the MnO$_2$-ZrO$_2$ dual oxide catalyst at a high temperature. Significant enhancement of catalytic activity was checked with increasing CO$_2$/EB ratios [52]. A TiO$_2$-ZrO$_2$ catalyst was used, and the proportion of TiO$_2$/ZrO$_2$ determined the catalytic activity [53–55,65]. A 60% titania content resulted the best performance for the ODEB [65,66]. Commercial Fe-supported catalysts are unsuitable for the oxidative dehydrogenation of ethyl benzene in the existence of carbon dioxide due to the atomization of the active catalytic site [56]. However, the use of appropriate dopants' support materials might enhance the activity by promoting re-oxidation of Fe$^{2+}$ and preventing coke deposition [57,58]. High product yield stability was observed in a mesoporous silica COK12-assisted CoO$_3$ catalyst [59]. The performance of several effective catalysts in the oxidative dehydrogenation of ethyl benzene to styrene in the existence of CO$_2$ is shown in Table 7.

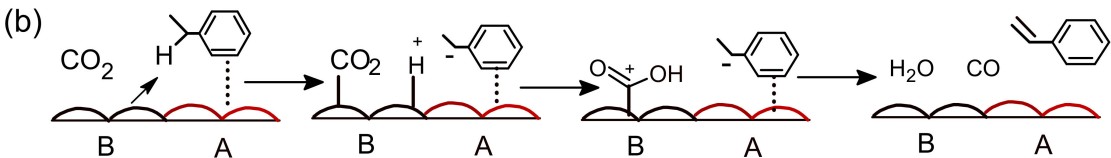

**Figure 4.** The procedure of oxidative dehydrogenation of ethylbenzene to styrene (**a**) without CO$_2$ and (**b**) with CO$_2$. (Reproduced with permission from [62]; copyright (2016), Elsevier).

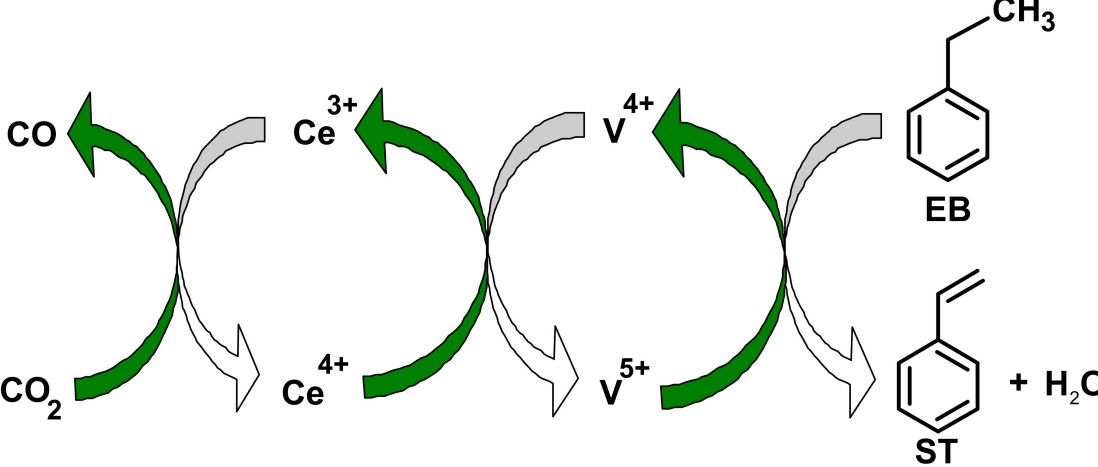

**Figure 5.** Red-ox cycle for CO$_2$-ODEB over the ceria-promoted vanadium catalyst. (Reproduced from [48]; copyright (2011), WILEY-VCH).

**Table 7.** Performance of $CO_2$ on oxidative dehydrogenation of ethylbenzene to styrene.

| Catalyst | Reaction Temperature (°C) | EB Conversion (%) | ST Selectivity (%) | ST Yield (%) | Ref. |
|---|---|---|---|---|---|
| $Co_3O_4$/COK-12 | 600 | 57.5 | 95.5 | 54.9 | [59] |
| $CeZrO_{4-\delta}$ | 550 | 7 | 97 | 6.8 | [61] |
| Na X zeolite | 545 | 9.4 | 89.6 | 8.4 | [60] |
| K X zeolite | 545 | 10.5 | 92.1 | 9.6 | [60] |
| $VO_x$/Al MCM-41 | 550 | 52.3 | 96.7 | 50.6 | [51] |
| $TiO_2$-$ZrO_2$ | 600 | 69.3 | 96.2 | 66.6 | [67] |
| $V_2O_5$/$SiO_2$ | 550 | 50.5 | 96.8 | 48.8 | [64] |
| $SnO_2$-$ZrO_2$ | 600 | 61.1 | 97.6 | 59.6 | [68] |
| $MnO_2$-$ZrO_2$ | 600 | 51.1 | 99.1 | 50.9 | [69] |

*3.2. Performance of $CO_2$ on Dehydrogenation of Ethane*

Ethylene is one of the most prominent raw materials in the chemical industry. Presently, it is used to produce industrial products such as PVC, ethylene glycol, ethylbenzene, ethylene oxide, and vinyl acetate. Commercially, ethylene is formed by steam cracking dehydrogenation of hydrocarbons and fluid catalytic cracking (FCC). These conventional methods have several major limitations including the reaction endothermicity, thermodynamic drawbacks, rapid coke formation, and high energy consumption. The oxidative dehydrogenation of ethane (ODHE) to ethylene in the presence of $CO_2$ as a mild oxidant is an environmentally friendly alternative method for the production of ethylene. A Cr-oxide catalyst with zeolite support was successfully used for the oxidative dehydrogenation of ethane in the presence of $CO_2$ as a soft oxidant. A novel Clinoptilolite-based Cr-oxide (Cr/CLT-IA) catalyst for the ODHE in the existence of $CO_2$ was investigated by Rahamani et al. [70]. This Cr-supported catalyst exhibits high selectivity and catalytic activity which was expected due to its acidity. Homogeneous, tunable smaller Clinoptilolite-based Cr catalyst particles with higher surface area can be generated. Thus, using a Cr/CLT-IA nano-catalyst may be feasible and favorable for the oxidative dehydrogenation of ethane to ethylene in the existence of $CO_2$. Cr/H-ZSM-5 ($SiO_2$/$Al_2O_3$ ≥ 190) outperformed the $SiO_2$-based catalyst in the oxidative dehydrogenation of ethane to ethylene with $CO_2$ [71]. $CO_2$ is a promising soft oxidant for the ODHE reaction acting as a channel for transporting heat to the endothermic dehydrogenation. Further, $CO_2$ improves the conversion by modifying alkanes and maintains the catalytic activity by eliminating coke from the catalyst surface. the texture of the Cr active sites and the catalyst activity are determined by the $SiO_2$/$Al_2O_3$ ratio. The presence of more alumina amount in the zeolite negatively affected the activity of the catalyst, due to the incorporation of alumina with the Cr into the catalyst structure, affecting the red-ox properties of Cr. Mimura et al. [71] reported on the dehydrogenation of ethane on a Cr-doped HZSM-5 catalyst which is established on the redox phase of the eminent oxidation type Cr species. In their work, $C_2H_6$ was absorbed on the acidic site of $CrO_x$ and H-ZSM-5. Then, the activated $C_2H_6$ reacted with $CrO_x$ (active O species) to produce ethylene. The $CrO_{x-1}$ species is then re-oxidized by the soft oxidant $CO_2$ regenerating the active O species and eliminating coke from the surface of the catalyst. The catalytic performance of the Cr-supported mesoporous catalyst, as well as a Cr-doped silicate MSU-1 catalyst, in the ethane oxidative dehydrogenation to ethylene in the presence of $CO_2$ was reported on by Liu et al. [72]. They initially observed high catalytic activity due to the Cr(VI) active species. However, even in the existence of $CO_2$, the reduction of Cr(VI) to Cr(III) occurred, resulting in the deactivation of the catalyst during the dehydrogenation reaction. Shi et al. [73] reported that Cr-supported Ce/SBA-15 catalysts were comprised of hexagonally ordered mesoporous frameworks and exhibited high catalytic activity in the oxidative dehydrogenation of ethane in the existence of $CO_2$. They confirmed the addition of Ce species using high-angle XRD, which increased the Cr species distribution in the Cr-Ce based SBA-15 zeolite. TPR results determined that Cr species in SBA-15-type zeolites are $Cr^{6+}$ and $Cr^{3+}$ groups. Among those two ions, $Cr^{6+}$ exhibited significant activity for the oxidative dehydrogenation reaction in the existence of $CO_2$. Including a Ce-supported in 5Cr/SBA-15 catalysts modified the red-ox properties and enhanced the activity of the catalyst. Ethane conversion was 55% and ethylene

selectivity was 96% on the 5Cr-10Ce/SBA-15 catalyst in the existence of $CO_2$ (Table 8). $Cr^{6+}$ is reduced to $Cr^{3+}$ during the oxidative dehydrogenation method reaction, however, in the presence of $CO_2$, $Cr^{3+}$ is re-oxidized to $Cr^{6+}$. $Cr_2O_3/ZrO_2$ supported catalysts with Fe, Co, Mn was also investigated in an effort to fully understand the excellent catalytic activity for the ethane dehydrogenation reaction to ethylene under $CO_2$ treatment [64,65,74]. The $Cr^{6+}/Cr^{3+}$ red-ox cycle is crucial in the oxidative dehydrogenation reaction, as is a $Fe^{3+}/Fe^{2+}$ red-ox cycle which was removes $H_2$ from the lattice oxygen. An SBA-15-based, Cr-modified catalyst using a Fe-Cr-Al alloy [75] also exhibited remarkable selectivity of ethylene and high ethane conversion in the oxidative dehydrogenation reaction with $CO_2$. Wang et al. [64] observed the red-ox properties and the acidity/basicity of the Cr-supported catalyst in the oxidative dehydrogenation of ethane to ethylene with $CO_2$. They found that Cr-supported catalysts exhibited different activities in the ODHE with $CO_2$. $Cr_2O_3/SiO_2$ showed higher ethane conversion and ethylene selectivity. The catalytic activities were ranked as follows $Cr/SiO_2 > Cr/ZrO_2 > Cr/Al_2O_3 > Cr/TiO_2$ [76,77]. Notably, $Cr_2O_3$ interacted more with $Al_2O_3$ than with $SiO_2$, resulting in tetrahedral $Cr^{6+}$ sites and declining activity [78]. Cr is one of the vital elements of various types of nano-catalysts (Table 9). The active site of these catalysts contains both $Cr^{3+}$ and $Cr^{6+}$. The $Cr^{6+}/Cr^{3+}$ ratio strongly influences the reducibility of Cr/H-ZSM-5 catalysts. The red-ox performance of Cr-supported catalysts is crucial for the oxidative dehydrogenation of ethane to ethylene in the presence of $CO_2$ as a soft oxidant. $Cr^{6+}$ (or $Cr^{5+}$) sites are reduced to $Cr^{3+}$ as ethane is dehydrogenated. Then, the reduced $Cr^{3+}$ sites are re-oxidized by carbon dioxide treatment. Mimura et al. reported that the highly active Cr-based catalysts had $Cr^{6+}$ or $Cr^{5+}$ species on the surface of the catalyst [79]. Apart from Cr-supported catalysts, several other effective catalysts have been used in research on ethane oxidative dehydrogenation. Among these, the Ni-Nb-mixed oxide catalyst performed very well at relatively low temperatures [80–82]. Additionally, a $TiO_2$-based Ga catalyst proved applicable for oxidative dehydrogenation with $CO_2$ [83].

**Table 8.** Catalytic activity for the dehydrogenation of ethane (Reproduced from [73]; copyright (2008), Springer (Berlin, Germany)).

| Catalyst | In the Presence of $CO_2$ | | | | In the Presence of Ar | | |
|---|---|---|---|---|---|---|---|
| | Conv. (%) | | Selectivity (%) | | Conv. (%) | Selectivity (%) | |
| | $C_2H_6$ | $CO_2$ | $C_2H_4$ | $CH_4$ | $C_2H_6$ | $C_2H_4$ | $CH_4$ |
| SBA-15 | 2.7 | 0.04 | 93.5 | 6.5 | 2.4 | 93.0 | 7.0 |
| 2.5Cr/SBA-15 | 39.6 | 15.9 | 95.5 | 4.5 | 30.2 | 89.7 | 10.3 |
| 5.0Cr/SBA-15 | 46.3 | 16.6 | 94.7 | 5.3 | 34.1 | 91.4 | 8.6 |
| 7.5Cr/SBA-15 | 45.3 | 18.8 | 92.2 | 7.8 | 33.9 | 92.8 | 7.2 |
| 10Cr/SBA-15 | 44.2 | 18.9 | 92.0 | 8.0 | 31.2 | 90.9 | 9.1 |
| 5Cr-5Ce/SBA-15 | 48.4 | 17.9 | 96.4 | 4.6 | 35.8 | 87.6 | 12.4 |
| 5Cr-7.5Ce/SBA-15 | 50.0 | 20.9 | 96.0 | 4.0 | 37.9 | 88.2 | 11.8 |
| 5Cr-10Ce/SBA-15 | 55.0 | 21.9 | 96.0 | 4.0 | 40.8 | 83.1 | 16.9 |
| 5Cr-15Ce/SBA-15 | 52.2 | 21.2 | 95.5 | 4.5 | 40.1 | 82.4 | 17.6 |

Reaction conditions: GHSV = 3600 mL/g h, T = 700 °C.

**Table 9.** Influence of $CO_2$ on oxidative dehydrogenation of ethane.

| Catalyst | Ethane Conversion (%) | Ethylene Selectivity (%) | Ethylene Yield (%) | Ref. |
|---|---|---|---|---|
| $Cr_2O_3$ (5 wt.%) CLT-IA | 39.7 | 98.8 | 39.2 | [70] |
| 3Cr/NaZSM-5-160 | 65.5 | 75.4 | 49.3 | [84] |
| $Cr_2O_3/Al_2O_3$-$ZrO_2$ | 36.0 | 56.2 | 20.2 | [85] |
| Cr/MSU-1 | 68.1 | 81.6 | 55.6 | [72] |
| $Cr_2O_3/ZrO_2$ | 77.4 | 46.3 | 35.8 | [86] |
| 2.5 Cr/SBA-15 | 46.3 | 94.7 | 43.8 | [75] |
| 5 Cr-10Ce/SBA-15 | 55.0 | 96.0 | 52.8 | [73] |
| 5% $Cr_2O_3/Al_2O_3$ | 19.2 | 56.5 | 10.8 | [77] |

Step-1

$$H_3C—CH_3 + CrO_x \rightleftarrows H_2C=CH_2 + H_2O + CrO_{x-1} \text{ (oxidative dehydrogenation)}$$

Step-2

$$H_3C—CH_3 \rightleftarrows H_2C=CH_2 + H_2 \text{ (Simple dehydrogenation)}$$

$$CrO_x + H_3C—CH_3 \rightleftarrows CH_4 + C + H_2O + CrO_{x-1} \text{ (methane and coke formation)}$$

$$H_3C—CH_3 + H_2 \rightleftarrows 2CH_4 \text{ (hydrocracking)}$$

Step-3

$$CrO_{x-1} + CO_2 \rightleftarrows CrO_x + CO \text{ (reoxidizing)}$$

$$C + CO_2 \rightleftarrows 2CO$$

### 3.3. Influence of $CO_2$ on the Alkylation of Toluene Side-Chain

The dehydrogenation of ethylbenzene produces the most styrene using the Friedel-Crafts alkylation reaction [87]. However, the ethylbenzene dehydrogenation method has some limitations such as catalyst deactivation and, high energy consumption [88,89]. Alkylation of the toluene side-chain is a promising alternative process that uses basic catalysts for the formation of styrene in the existence of $CO_2$. Another process was reported by Sindorenko et al. [90] utilizing $K^+$ and $Rb^+$ ion transposing Faujasite supported catalysts in 1967. However, the catalytic conversion of toluene and styrene monomer (SM) selectivity was low (Table 10) [91]. The side-chain alkylation is primarily carried out on solid base catalysts [92–96]. Toluene side-chain alkylation with methanol enhanced by the promotional use of alkali metal oxides. Greater catalyst acidity accelerates methanol dehydration, [97] while low concentrations of alkali metal ions prevent the decomposition of formaldehyde produced from methanol [98]. Thus, catalysts for this reaction must be optimized for their acidity and basicity [99]. Generally, catalyst sites for the side-chain alkylation are limited to alkali metal-altered zeolites [100]. One reliable, widely studied catalyst is the cesium ion-exchanged or $Ce_2O$-impregnated zeolite-X. The advantages of a MgO-supported mesoporous catalyst for this reaction has also been reported by Park el al. [101]. Hattori et al. observed that the impregnation of $Cs_2O$ in ion-exchanged zeolite-X results in high conversion of toluene, owing to the strongly basic sites [102]. Carbon dioxide has been under consideration as a renewable, low-cost, safe, and environmentally beneficial feedstock in current years. $CO_2$ utilization is difficult for commercial applications, owing to its high thermal stability as well as the solid oxidation phase [103]. Hence, remarkable research efforts are being directed to detect innovative technologies for the utilization of $CO_2$. Toluene side-chain alkylation was performed to assess the efficacy of the catalytic approach with methanol over cesium-supported catalysts. Toluene and methanol conversion over the Cs-X and Cs-modified zeolites in the presence of He and $CO_2$ are shown in Table 10. In these reactions, styrene and ethylbenzene were formed as main products. Other side-chain alkylated components, including cumin and $\alpha$-methyl styrene, as well as other xylenes, tri-methylbenzene, and benzene were identified as by-products. When the catalytic reaction was carried out in the existence of $CO_2$, methanol and toluene conversion increased. Though the styrene selectivity decreased, there was a significant increase in the conversion as well as product selectivity in the presence of He and $CO_2$ streams. TG/DTA analysis of the used Cs-X catalyst in the presence of $CO_2$ and He streams is shown in Figure 6. In the range of 25–200 °C, weight loss occurred owing to the desorption of adsorbed water [94]. The continued weight loss in the 200–450 °C region occurred due to the deposition of coke on the surface of the catalyst. Relatively high quantities of coke were deposited on the Cs-X catalyst in the existence of the $CO_2$. This suggests greater deactivation of the catalyst in the presence of carbon dioxide owing to coke deposition [89]. Still the Cesium-supported catalysts performed better in the presence of $CO_2$ than under He in terms of toluene and methanol conversion. $CO_2$ acted as a significant performance in hydrogen skulking and enhanced the reaction

rate in the decisive route. Additionally, $CO_2$ increases alkylation to produce cumin and α-methyl styrene, which are side-chain alkylation products. Further, the increased toluene conversion enhances the aromatic yields.

**Table 10.** Performance of $CO_2$ in the toluene side-chain alkylation (Reproduced with permission from [91]; copyright (2018), Elsevier).

| Catalyst | Carrier Gas | MeOH Conv. (%) | Toluene Conv. (%) | Selectivity (%) | | |
|---|---|---|---|---|---|---|
| | | | | SM | EB | Others |
| Ce-X | He | 12.54 | 1.42 | 78.61 | 15.32 | 6.07 |
| | $CO_2$ | 35.35 | 3.48 | 45.83 | 33.36 | 20.81 |
| $Cs_2O/$ | He | 46.48 | 3.59 | 28.76 | 68.02 | 3.22 |
| Cs-X | $CO_2$ | 39.16 | 2.52 | 36.02 | 43.02 | 20.78 |

Reaction conditions: WHSV = 2.1 h$^{-1}$, Reaction temperature = 425 °C, Toluene/MeOH molar ratio = 2, SM = Styrene Monomer and other byproducts = Cumene, Xylenes, TMB and Benzene.

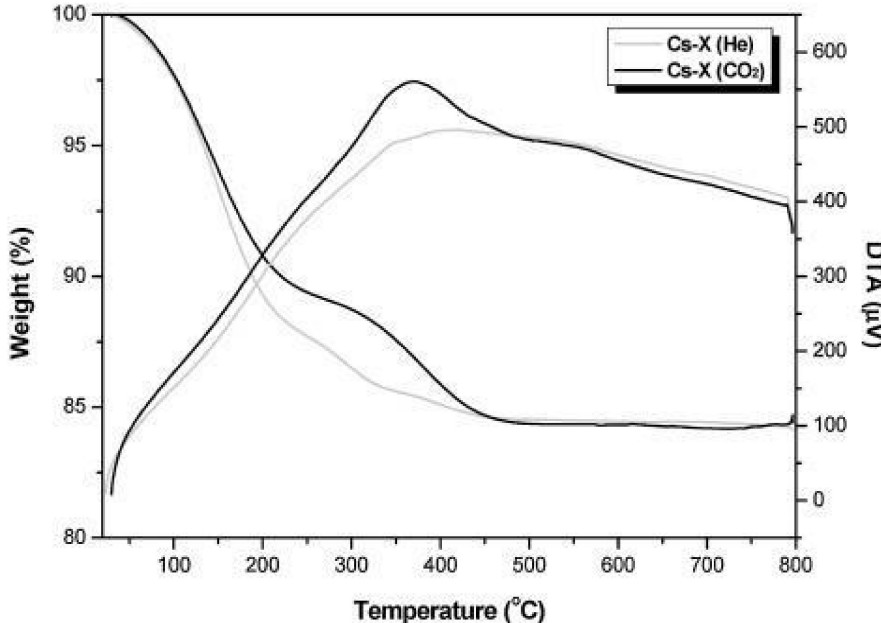

**Figure 6.** TG/DTA results obtained for used Cs-X catalyst in the presence of $CO_2$ and He streams (Redrawn with permission from [91]; copyright (2018), Elsevier).

### 3.4. Role of $CO_2$ on Dehydrogenation of Propane

Propylene is the most prominent raw material in the chemical industries. It is primarily manufactured by steam cracking and propane oxidative dehydrogenation [104–106]. Oxidative dehydrogenation (ODH) is preferred due to its low energy requirements and lack of thermodynamic limitations [107,108]. However, the ODH reaction with $O_2$ occurs under potentially flammable conditions and forms of carbon oxides due to over-oxidation with low selectivity [109,110]. This complication can be resolved using $CO_2$ as a mild, safer oxidant. Thus, this reaction is a favorable example of $CO_2$ utilization. Interestingly, $CO_2$ was used as a mild oxidant to shift the equilibrium more toward the products, as well as enhance the dehydrogenation over the coupling between propane oxidative dehydrogenation to propylene and the reverse water gas shift (RWGS) reaction [111–113]. Dehydrogenation of propane occurred on the acid site of the catalyst. The $SiO_2/Al_2O_3$ proportion is critical in determining both the catalyst physicochemical properties and its reactivity characteristics [114–117]. The HZSM-5, SBA-15, MCM-41, SBA-1 catalyst which is a two-dimensional microchannel system, has been used in the oxidative dehydrogenation of alkanes especially for the conversion of methane to propane in the existence of $CO_2$. Various research groups have reported

on the influence of catalyst acidity in the oxidative dehydrogenation reaction with $CO_2$. The activity of the zeolites decreased with increasing Si/Al proportion in HZSM-5 based $Ga_2O_3$, although the selectivity increased, as shown is in Figure 7 [118]. Lewis acidity is present in the metal oxide ($Ga_2O_3$) catalyst, while Bronsted acidity is present in HZSM-5. Thus, extracting the aluminum from HZSM-5 declines the Bronsted acidity more than it decreases the Lewis acidity. Several transition metals, such as vanadium, molybdenum, and chromium, have been used to support catalysts for ODH of light alkanes including propane [105,112,119,120]. Among these, chromium oxide provided high catalytic performance with $CO_2$, despite fractional deactivation by coke production. Chromium oxide enhanced propane conversion and the propylene selectivity by expelling $H_2$ produced in the ODH reaction [112]. The catalytic performance of Cr-supported catalysts was observed by the character of chromium categories on the support surface of the catalysts [121–125] Park et al. found that different Cr doping of Cr-TUD-1 catalysts (3, 5, 7 and 9 wt.%) with soft oxidant ($CO_2$) were formed by MW irradiation and investigated the propane oxidative dehydrogenation [126]. The effect of reaction temperature on the oxidative dehydrogenation of propane in the existence of $CO_2$ as a mild oxidant over the Cr-TUD-1 catalyst (7 wt.%) was investigated thoroughly to improve the catalytic activity. The conversion of $CO_2$ was 3.5% at 550 °C and improved to 5.5% at 650 °C. To demonstrate the importance of $CO_2$ in the propane oxidative dehydrogenation on Cr-TUD-1 catalysts, the process was carried out at 550 °C on 7 wt.% catalyst under the same conditions in the presence of $CO_2$ as well as He. The decline in the catalytic activity of the catalyst with helium may be due to coke production and the reduction of the Cr groups on the surface of the zeolite. The proposed mechanism of propane oxidative dehydrogenation over metal oxide surfaces with the $CO_2$ stream is shown below [112]:

A weak exclusive propane adsorption on the lattice oxygen

$$C_3H_8 + O^* \rightarrow C_3H_8O^* \tag{2}$$

C-H schism via H-abstraction from propane utilizing an abutting lattice oxygen

$$C_3H_8O^* + O^* \rightarrow C_3H_7O^* \tag{3}$$

Propylene desorption by hybrid expulsion from adsorbed alkoxide groups

$$C_3H_7O^* \rightarrow C_3H_6 + OH^* \tag{4}$$

Reconsolidation of OH groups to produce $H_2O$, reduced metal center (*)

$$OH^* + OH^* \rightarrow H_2O + O^* + {}^* \tag{5}$$

Re-oxidation of abridged M-centers by separating chemisorptions of $CO_2$

$$2CO_2 + {}^* + {}^* \rightarrow 2CO + 2O^* \tag{6}$$

To evaluate the deactivation of the catalyst by coke creation and the enhancement of $CO_2$, (Equation (7)) can be used as the deactivation parameter:

$$\textbf{Deactivation parameter (\%)} = \text{Conversion of propane (initial amount − final amount)/ (initial amount) * 100} \tag{7}$$

The rate of Cr degradation by $H_2$ liberated from dehydrogenation is faster than the rate at which $CO_2$ re-oxidizes the degraded Cr species, resulting in catalytic deactivation. Selective adsorption properties can be improved by surface functional groups on activated carbons. Thus, surface treatment of activated carbon may result in more selective and efficient adsorption of the gas, liquid and the alleviation of pollution [127].

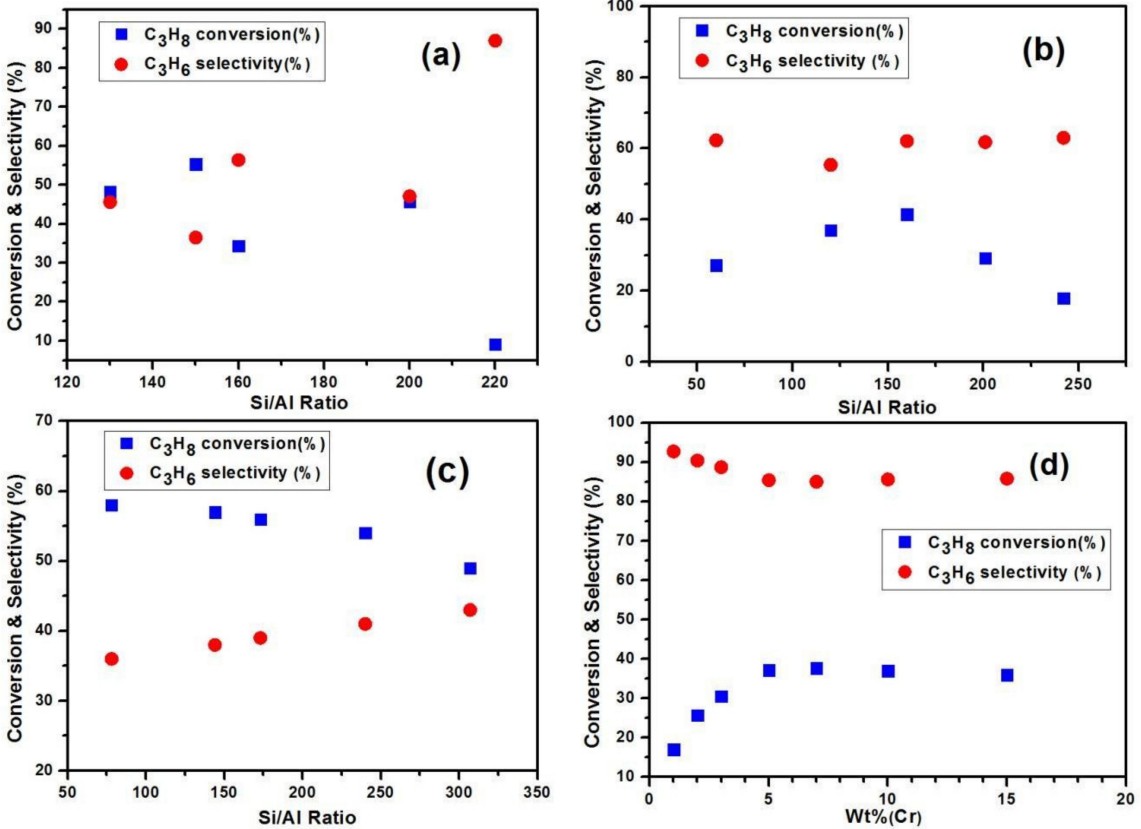

**Figure 7.** Influence of Si/Al proportion on the efficiency of (**a**) $Ga_2O_3$/ZSM-48 zeolites (Reproduced from [118]; copyright (2012), Elsevier), (**b**) ZnO-HZSM-5 zeolites in the oxidative dehydrogenation of propane along with $CO_2$ (Reproduced from [105]; copyright (2009), Elsevier), (**c**) $Ga_2O_3$/M-HZSM-5 zeolites in the absence of $CO_2$ (Reproduced from [118]; copyright (2012), Elsevier), (**d**) Influence of Cr substance on the effectiveness of Cr/SBA-15 in the carriage of $CO_2$ (Reproduced from [128]; copyright (2012), Elsevier).

## 4. Conclusions

This review article has comprised a number of $CO_2$ conversions, which are still in the research scale. These promising technologies are mitigating the continuously increasing atmospheric $CO_2$ concentration. Among the methods employing $CO_2$, the ethyl benzene ODH process has seen significant progress. Currently, most of the ethylbenzene dehydrogenation plants apply the oxidative dehydrogenation method, which leads to large heat losses upon compression at the gas–liquid separator. Further, this reaction is thermodynamically restrictive and energy intensive. Several industrial companies such as SABIC (Saudi Basic Industry Corporation, Saudi Arabia), Samsung General Co. in south Korea have tested the catalytic consummation for this method. The commercial implementation of such a process may support the economics of styrene monomer production. According to European Rubber Journal (ERJ), Asahi Kasei Chemical Company 's (Japan) 6th generation SBR (Styrene-butadiene rubber) is currently being tested by many customers in the world with positive feedback and company is planning to commercialize some grades in 2021. Moreover, Trinseo's highly functionalized SPRINTAN$^{TM}$ 918S Solution-Styrene Butadiene Rubber (S-SBR) has awarded second position in the elastomers for sustainability initiative of the European Rubber Journal. Based on lab indicator data confirmed by tire customers, grade 918S (compared to non-functionalized high-grip SSBR) improves fuel efficiency of the whole car approximately 1.5%. Considering in Europe alone, the benefit of this increased fuel efficiency would translate in approximately 540 tons less fuel consumed or a reduction of $CO_2$ emissions by 1.3 million tons.

Several methods using $CO_2$ as a mild oxidant have appeared in the technology sector. It is a long-term goal and alluring dream for chemical engineers to establish commercial industries based on the utilization of $CO_2$. Challenges for the commercial utilization of this technology include the process rate required to ensure $CO_2$ conversion with low coke deposition, the need to decrease energy expenditure, and the need for improved catalysts offering higher conversion. Despite the challenges, there is great room for catalyst improvement in these sectors. Recently, the carbon XPRIZE is a $20 million competition to capture and $CO_2$ conversion which is jointly funded by COSIA (Canada's Oil Sends Innovation Alliance) [129]. Most of the countries' governments are concern about climate changes with a high priority. China, the world's largest energy consumer and carbon emitter, announced USD 360 billion in renewable energy investments by 2020 to reduce carbon emissions [130]. Canada has implemented federally a carbon pricing policy with a current tax of USD 10/ton $CO_2$ and a steady rise to USD 50/ton $CO_2$ nationwide by 2022. However, the positive effects of $CO_2$ in benzene hydroxylation over commercial and hierarchical zeolites in the liquid phase as well as the gas phase are under investigation by our group, wherein the byproducts are various aromatic compounds. The recycling of $CO_2$ from the atmosphere to fuels, chemicals will lead to a real sustainable future for humanity. We expect that the use of $CO_2$ as a promoter and as a mild oxidant at the laboratory level can be translated to the industrial scale in the future, thus contributing also to the world economy.

**Author Contributions:** S.T.R.: Writing original draft; J.-R.C.: Editing; J.-H.L.: Editing; S.-J.P.: Writing review & editing. All authors have read and agreed to the published version of the manuscript.

**Funding:** This work was supported by the Technology Innovation Program (or Industrial Strategic Technology Development Program-Development of technology on materials and components) (20010881, Development of ACF for rigid (COG)/ flexible (COP) and secured mass production by developing core material technology for localizing latent hardener for low temperature fast curing) funded By the Ministry of Trade, Industry & Energy (MOTIE, Korea) and supported by Korea Evaluation institute of Industrial Technology (KEIT) through the Carbon Cluster Construction project [10083586, Development of petroleum based graphite fibers with ultra-high thermal conductivity] funded by the Ministry of Trade, Industry & Energy (MOTIE, Korea).

**Conflicts of Interest:** The authors declare that they have no conflicts of interest.

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
