# Peer review of "The Role of CO2 as a Mild Oxidant in Oxidation and Dehydrogenation over Catalysts: A Review"

_catalysts, doi:10.3390/catal10091075_

Round 1

Reviewer 1 Report

Dear Authors,

The Authors analyzed many papers and based on them, they presented their own study. The idea of the article is based on the necessity to reduce CO2 emitted to the atmosphere. The authors show the possibility of using CO2 in catalytic oxidation reactions. However, in order to show how much (and not only in terms of research, because it is a marginal scale) you can influence the use of CO2, in a significant amount for the environment, it should be shown quantitatively. Industrial technologies in which the described catalytic processes will translated into a technical scale should be indicated. How much CO2 in such technologies will be used and how much it will reduce CO2 emissions to the atmosphere. Indicating only the direction of CO2 use in research, is definitely not enough to conclude that it will be of economic importance.

 In relation to all chapters:

The Authors do not state, whether the data they present were obtained on a laboratory scale or on a fractional technical or technical scale. It must be clearly defined whether the presented results of the work were obtained in basic or application research. If the results come from laboratory tests, it is important to indicate the mass of the reagents - was it only a micro scale, or a gram or a kilogram scale.

It should be specified under which process conditions the oxidation reactions with CO2 take place? You cannot compare solutions without specifying the process temperature, pressure, etc.

Conclusions

Conclusions should be reedited. Conclusions should be more developed and result directly from the previously presented content. They should not be in the form of a discussion. No other papers should be cited in the conclusions.

The sentence (verse 61-63) is incomprehensible:

Because it is softer than O2 and H2O, CO2 escapes the burning of important hydrocarbons. CO2 is less precarious than N2O and  SO2 [18].

There are unnecessary repetitions in one chapter: (lines 85-86)

CO2 behaves as a soft oxidant in oxidative dehydrogenation (ODH) reactions and offers several advantages in comparison to alternative oxidants.

Reviewer 2 Report

Reviewer´s comments:

Review „The role of CO2 as a mild oxidant in oxidation and 2 dehydrogenation over catalysts: A review“ of the authors team Sheikh Tareq Rahman, Jang-Rak Choi, Jong-Hoon Lee and Soo-Jin Park provides quantity of very interesting informations dealing with non-conventional utilization of CO2 as diluent and mild oxidant for oxidative dehydrogenation type reactions.

However, this version of manuscript requires major revisions dealing with improvement of text formulations for clarifying presented information to the readers.

Introduction: Line 25: „CO2 is utilized in the power generation…“ Do you mean that CO2 is produced as the undesirable combustion product?

Line 51: „…absorption of hydrogen from alkanes,…“ Do you mean chemisorption (by oxidation)?

Line 52: „…using CO2 as a catalyst to create CO and oxygen species…“ CO2 takes part in these described processes in role of the catalyst or reactant?

Line 60-61: „CO2 offers various benefits as a mild oxidant over other oxidizing …., N2O and CO2“ Please, improve this sentence.

Line 64-65: Please, add references to this statement „Additionally, CO2 is used as a carbon source in the decoking process (C + CO2 = 2CO) which sustains catalytic activity.“

Line 79: Please, add the reference number into the sentence: „Bartholomew et al. studied the oxidizing capability of different gases in the…“

Chapter 2:

Line 89: It would be very useful to focus readers on the reaction scheme 3 describing the cyclohexene oxidation by CO2 even in the first sentence of this chapter.

Please, define „Psi“ unit used in Table 1 (line 107) or use SI units.

In some cases, references could be added into the text, concretely: line 35 „Biomass conversion also utilizes CO2.“ [?]

Line 137: „cyclo-dodecane“ or cyclododecene?

Line 150: Table 2: What means abbreviation Ocand O?

Line 154: Scheme 4: Please, improve the scheme 4 (benzaldehyde is mentioned in heading, however, phenylglyoxylic acid is ilustrated in this scheme).

Lines 156-163: In subchapter focused on p-xylene oxidation the oxidation of styrene is discussed. Please, re-organize the text.

Line 186: Sentence: „No carbon dioxide was formed by burning p-xylene over the catalyst at 375 ℃; however…“

Line 194: Table 4: Please, define exactly structure of „Trimethyl biphenyl methane“

Line 308: Please, clarify the sentence:  The oxidative dehydrogenation of ethane (ODHE) to ethylene in the existence of CO2 as a mild oxidant… Do you mean co-action? (the same on line 311)

Line 376: Please, clarify the sentence:  However, the catalytic conversion of toluene and styrene selectivity was low [84].

Conclusion:

Line 478: Please, control the importance of Reference 113:

„Burri, A.; Hasib, M.A.; Mo, Y.H.; Reddy, B.M.; Park, S.-E. An Efficient Cr-TUD-1 Catalyst for Oxidative 763 Dehydrogenation of Propane to Propylene with CO2 as Soft Oxidant. Catal. Letters 2018, 148, 576–585.“

for oxidative dehydrogenation of ethylbenzene to styrene described in sentence on lines 476-478.

Line 490: reference is completely missing in the sentence: „effects of CO2 in benzene hydroxylation over commercial and hierarchical zeolites in the liquid phase…“

Generaly:

In many cases, the references mentioned in the article are incomplete. Please, add numbers of cited works to the name of the mentioned first author (in line: 121 („Iwata et al“ [?], line 157: Aresta et al[?], line 163: Park et al[?], line: 221: Zhang et al[?], line 333: Shi et al[?], line 349: Wang et al[?], …375 (Sindorenko [please, add number of reference], line 384 Park et al [?], line 484: G.K. Surya Prakash et al [?])

Abbreviations should be explained:

Line 116: DBH

Line 126: CN

Line 148: NHC

Line 176: CVD

Line 232: GHSV

Line 256: ODH

Line 267: ODHEB

Line 272: TZ

Line 273: CO2-ODEB

Line 280: AIMCM

Line 315: CLT-IA

Round 2

Reviewer 1 Report

Dear Authors,

I sustain my previous review. You did not respond my remarks, that I have included in my previous review. You did not include any explanation. 

Reviewer 2 Report

The submitted new version of manuscript is markedly improved, however, still few faintnesses or errors should be corrected. 

  1. Please, consider the phrase (page 1, line 26) "CO2 is utilized in the power generation...", utilized or for example emitted?
  2. Please, correct on page 1, line 34, end of the sentence: "...other gases.[5]"
  3. Page 4, line 113: "...oxidation of cyclohexane..." OR CYCLOHEXENE?
  4. The same, on page 5, line 156: "...and cyclododecane (n =8)..." Do you really mean saturated cyclic hydrocarbon or this is a mistake or do you comment oxidation of cyclic olefin cyclododecene?
  5. Page 7, line 212: Please, consider the phrase: "It was observed that no carbon dioxide was formed by the burning of p-xylene..." By my best of knowledge the word "burning" means complete oxidation, in the case of xylene complete oxidation means production of CO2 and H2O and the above discussed phrase miss point.
  6. Page 8, lines 232 and 234: the Reference number are still missing.
  7. Page 12, line 326 and 327: abbr. "AIMCM-41" OR Al MCM-41 (used in Table 7)?
  8. Page 19, line 526: "According to drisdell group,..." OR Drisdell group?

Round 3

Reviewer 1 Report

The authors made corrections and additions to the article. However, they did not take into account all my comments. It seems that the authors operate only at the level of laboratory research, without deepening the knowledge of translating technology into an industrial scale.

In my opinion, the additions included in the conclusions are not conclusions, but since other reviewers accept it, I agree to this version. However, I believe that its proper preparation would add value to the manuscript.

Editorial Note:

After the amendments were made, the order of the citation numbers in the text was disturbed.
